# A Thermodynamic Approach to Measuring Entropy in a Few-Electron Nanodevice

**DOI:** 10.3390/e23060640

**Published:** 2021-05-21

**Authors:** Eugenia Pyurbeeva, Jan A. Mol

**Affiliations:** School of Physics and Astronomy, Queen Mary University of London, Mile End Road, London E1 4NS, UK

**Keywords:** nanoscale system, quantum transport, Coulomb blockade, entropy measurement, thermodynamic relations

## Abstract

The entropy of a system gives a powerful insight into its microscopic degrees of freedom; however, standard experimental ways of measuring entropy through heat capacity are hard to apply to nanoscale systems, as they require the measurement of increasingly small amounts of heat. Two alternative entropy measurement methods have been recently proposed for nanodevices: through charge balance measurements and transport properties. We describe a self-consistent thermodynamic framework for applying thermodynamic relations to few-electron nanodevices—small systems, where fluctuations in particle number are significant, whilst highlighting several ongoing misconceptions. We derive a relation (a consequence of a Maxwell relation for small systems), which describes both existing entropy measurement methods as special cases, while also allowing the experimentalist to probe the intermediate regime between them. Finally, we independently prove the applicability of our framework in systems with complex microscopic dynamics—those with many excited states of various degeneracies—from microscopic considerations.

## 1. Introduction

Entropy is one of the cornerstones of thermodynamics. Boltzmann’s original insight in his namesake equation $S=k_{B}\ln\Omega$ summarises the main source of power of thermodynamics—the ability to connect a macroscopic quantity to the number of microstates Ω in a system-independent way.

In the macroscopic realms of thermodynamics, large Hamiltonian systems with many degrees of freedom, the number of accessible microstates is so great that the microscopic meaning of entropy is largely ignored, while it is treated as a state function dependent on other, more readily measured, state functions. As the size of the system and, with it, the volume of its state-space are reduced, individual microstates come into focus and the knowledge of entropy can provide information about the number and relative probabilities of the microstates of the system in question. Entropy measurements have been performed in various microscopic systems: spin-ice [1], 2D electron gas in GaAs structures [2,3,4] and fractional quantum Hall states [5,6]. Yet, as experimentally accessible thermodynamic systems become progressively smaller, from quantum dots [7,8] and quantum dot systems, through molecules [9,10,11] to single atoms [12,13] and individual electron spins [14], the usual approach to entropy measurement, based on the Clausius definition dS=δQ/T becomes increasingly difficult since it involves measuring ever-decreasing heat flows. Therefore, the problem of finding an alternative entropy measurement method applicable for small systems presents itself.

Recently, two such methods were developed to measure the entropy of few-electron nanodevices. The first method relies on measuring the charge state of the nanodevice [15,16], while the second is concerned with the electronic transport through the device [8,17,18]. Here, we explore the application of thermodynamic relations to few-electron systems and show that both direct entropy measurement methods are special cases of a more general relation between the average electron occupation of the nanodevice and its entropy. We will derive this relation from purely thermodynamic considerations, i.e., without any knowledge of the microscopic details of the nanodevice.

This paper is organised as follows: first, in Section 2, we discuss the system under consideration and the parameters characterising it. Next, in Section 3, we look into the effects of degeneracy arising from the rate equation and how these have been used previously to measure entropy [15]. Then, in Section 4, we discuss the subtleties of correct thermodynamic treatment of devices with significant particle number fluctuations and the parameters that need to be employed and derive a thermodynamic relation for a system without excited states. We show that it describes both previously used entropy measurement methods [15,17] as special cases, before expanding the approach to more complex systems with multiple excited states in Section 5. Finally, we conclude with a brief summary in Section 6.

## 2. The System

Following the previously described electric entropy measurement methods [15,17], we will focus on single-electron nanodevices such as quantum dots [7,8] and single-molecule devices in the resonant transport regime (sequential tunnelling) [19]. Experimental measurements of these devices fall into two broad categories, as shown in Figure 1: charge state measurements [15]; and transport measurements (including thermoelectric transport) [8,17,18]. The free parameters in both experimental setups are the temperatures of the baths and the energy level of the quantum dot (or molecule, we will refer to both as the quantum dot in the future). In the transport measurement setup (Figure 1b), additional degrees of freedom are the temperature difference between the baths and the bias voltage; however, we will look at the quasistatic case where both are infinitesimally small.

We will consider the case where the quantum dot has only two energetically accessible charge states, with *N* and N+1 electrons occupying it, and define the single-particle energy of the quantum dot ε=E(N+1)−E(N) as the energy difference between the total energy of the quantum dot in the N+1 and *N* charge states [20]. The single-particle energy level can be controlled by applying a gate voltage $V_{g}$, $\varepsilon =\varepsilon_{0}-e\alpha V_{g}$, where the lever arm α is given by the electrostatic coupling between the gate and the quantum dot. For now, we will forgo the consideration of excited states and assume that energy depends on the charge state only. See Section 5.1, for the discussion of ε in case of energy splitting of a charge state.

We treat the electrodes as ideal thermal baths with Fermi-distributions and chemical potential μ. For all practical applications up to and above room temperature, the Fermi-gas in the electrodes remains highly degenerate; therefore, we can put $\mu =E_{F}$ and neglect its dependence on temperature. Since the quantum dot and the electrodes are in equilibrium with respect to particle exchange (arbitrarily close to equilibrium in the transport measurement setup), the chemical potential of the of the quantum dot is equal to μ.

We emphasize that it does not imply that ε and μ can be equated, as suggested by Hartman et al. [15]. The single-particle energy level ε, the additional energy the quantum dot gets when entered by a new electron, is often referred to as an electrochemical potential (see, for example [21]); however, it is not one from a thermodynamic perspective, and its use in this capacity is, in the general case, incorrect. By definition, $\mu ={\left(\partial U/\partial N\right)}_{S,V}$, whilst in our case, an electron entering a quantum dot necessarily changes its entropy. Below, we will demonstrate that entropy can be measured directly from ε−μ.

## 3. Rate Equations

### 3.1. Degeneracy Effects in the Rate Equation

First, we consider the effects of transport level degeneracy emergent from a rate equation approach. Starting with a quantum dot coupled to a thermal bath, with only charge states accessible containing *N* and N+1 electrons, the hopping rates of the electrons to and from the dot are proportional to the degeneracies of the charge states [22]:(1)$$\left\{\begin{array}{c} \Gamma_{T}=\gamma d_{N+1}f\left(\varepsilon \right)\\ \Gamma_{F}=\gamma d_{N}\left[1-f(\varepsilon )\right] \end{array}\right.$$
where $\Gamma_{T/F}$ are the rate of electron hopping to/from the quantum dot, $f\left(\varepsilon \right)={\left(\exp\left[\left(\varepsilon -\mu \right)/k_{B}T\right]+1\right)}^{-1}$ is the Fermi-distribution of the bath, γ is a geometric rate factor, and $d_{N/N+1}$ is the degeneracy of the charge state with N/N+1 electrons.

As the probability for the system to occupy a given charge state, and with it the time it spends in each state depends on the hopping rates, the probabilities $p_{N/N+1}$ of each charge state occupation can be written as:(2)$$\left\{\begin{array}{c} p_{N}=\frac{\Gamma_{F}}{\Gamma_{T}+\Gamma_{F}}\\ \;\\ p_{N+1}=\frac{\Gamma_{T}}{\Gamma_{T}+\Gamma_{F}} \end{array}\right.$$
If the charge state of the system can be measured directly with high-enough time resolution, the fraction of the total time the system occupies a given charge state is equal to $t_{N}/t_{N+1}=p_{N}/p_{N+1}$ and contains information about relative charge state degeneracies [23].

In the simplest case, the transport level of a quantum dot has a two-fold degeneracy due to spin orientation, and the degeneracies $d_{N/N+1}$ depend on the parity of *N*—for an even *N*, the transport level is empty and an additional electron can have two possible spin orientations, while for an odd *N*, the transport level already contains one electron (with an arbitrary spin orientation), and an additional electron enters with an opposite spin. Therefore, $d_{N+1}/d_{N}=2$ for an even *N* and $d_{N+1}/d_{N}=1/2$ for odd *N*. The mean excess population of the quantum dot *n* is equal $p_{N+1}$ and varies between 0 and 1. See Figure 2a, for the dependence of *n* on ε for a non-degenerate level and two parities of *N* of a two-fold degenerate level.

A second charge state degeneracy effect is manifested in the quantum transport setup (Figure 1b). For a non-degenerate transport level, the conductance of the device is highest for the transport level coinciding with the chemical potentials of the electrodes (ε=μ)—see Figure 2c. The change in the hopping rates due to the level degeneracy causes a temperature-dependent shift in the peak conductance of a single-electron transistor predicted in [24] and experimentally measured in [8,18] (Figure 2a). For a two-fold degenerate transport level, standard for the spin-degeneracy of electronic current through a quantum dot $\left(\varepsilon_{p}-\mu \right)/k_{B}T=\pm \ln2/2$, where $\varepsilon_{p}$ is the value of ε corresponding to peak conductance and its sign depends on the parity of *N*—Figure 2c.

Both effects—the conductance peak shift and the charge state occupation probability depend on the degeneracies and, therefore, allow to construct an entropy difference between the charge states retroactively by extracting relative degeneracies. However, this is not a “true” entropy measurement, as it is based on assumptions about the hopping rates and in this form is only applicable to a single energy level with $d_{N/N+1}$ degeneracy in the weak coupling limit, while expansion to more complex systems, even a quantum dot in a magnetic field, is not possible, as entropy is artificially constructed utilising prior knowledge of the system.

In order for a method to be capable of measuring the entropy difference between the charge states with arbitrary dynamics (each charge macrostate can consist of a number of microstates with different energies), and for a method to be truly thermodynamic, it has to be free of any assumptions based on our knowledge of the system. One approach to these prior knowledge-independent, “direct” entropy measurements lies in applying the Maxwell relations to nanodevices.

### 3.2. Detailed Balance Approach to Maxwell Relations

The first alternative fully thermodynamic entropy measurement methods that did not involve the measurement of heat were developed for quantum Hall states [25,26] and utilised Maxwell relations to relate the derivative of entropy to other, more readily measurable parameters.

The idea proposed by Hartman et al. [15] was to apply a Maxwell relation to a quantum dot device:(3)$${\left(\frac{\partial \mu }{\partial T}\right)}_{N}=-{\left(\frac{\partial S}{\partial N}\right)}_{T}$$
which connects the change in entropy with the number of electrons, the quantity we are most interested in, with others, which can be measured directly, without making any previous assumptions about the nature of the system.

However, a quantum dot with few electrons is far from the usual system for application and Equation (Equation 3) has to be treated with utmost care.

One of the issues with the way the Maxwell relation was used in [15] was treating the right-hand side derivative as a ratio of finite increments ΔS/ΔN, where ΔN=1 since only one electron can tunnel in or out, and ΔS is the entropy change associated with a single tunnelling event. Since the quantum dot is a few-electron system and only two charge states are accessible, ΔN=1 is not only a large, but the only possible fluctuation. Moreover, under the treatment of *N* as the particle number in Equation (Equation 3), its left-hand side loses its meaning, since in all states, except the two extreme ones of ε−μ→±∞ the particle number constantly fluctuates between *N* and N+1 and cannot be taken as constant.

Due to the issues described above and the non-thermodynamic use of the chemical potential we mention in Section 3.1, the Maxwell relation can only serve as motivation in [15]. The correct derivation of the relationship between entropy and energy used in [15] follows from detailed balance—if the probabilities of finding the quantum dot in both charge states are equal (the point the authors look at experimentally), the tunneling rates Equation (Equation 1) in and out are equal, which results in the equation $d_{N+1}/d_{N}=\left(1-f\left(\varepsilon \right)\right)/f\left(\varepsilon \right)$, which after taking a logarithm takes the form:(4)$$\frac{\varepsilon -\mu }{T}=k_{B}\left(\ln d_{N+1}-\ln d_{N}\right)=\Delta S$$
This equation resembles the Maxwell relation written for the quantum dot; however, it is only valid for one value of ε−μ—the one corresponding to equal charge state probabilities, while a Maxwell relation holds true for all values of external parameters. This limitation is a further indication that the results in [15] do not follow directly from the Maxwell relation, while the derivation from the detailed balance limits the range application of the method to a single degenerate level.

Having described the difficulties in using the Maxwell relations in small, few-electron systems, in the following sections, we look at all the parameters involved separately, discuss their applicability and derive a thermodynamic relation that correctly describes a quantum dot.

## 4. Thermodynamic Relation, No Excited States

### 4.1. Derivation and Entropy Definition

First, we consider a preliminary case of a system where energy depends on the charge state of the quantum dot only—each charge state might have several microstates, but they all have the same energy, $E(N^{\prime })$, where $N^{\prime }$ is equal to *N* or N+1 and E(N+1)=E(N)+ε.

As thermodynamics operates with averaged quantities, to derive a *general* thermodynamic relation between entropy difference between the two charge states of a quantum dot and its energy level, we need to consider the mean population of the quantum dot $\bar{N}$. The single-particle energy level has a mean occupation *n* between 0 and 1, while the base population of the dot *N* remains unchanged. Since $\bar{N}=N+n$, the mean additional energy is εn and the mean free energy $\bar{F}=E\left(N\right)+\varepsilon n-TS$. As *N* remains constant, the derivatives in the Maxwell relation for the quantum dot can be taken by the mean excess population, which yields:(5)$${\left(\frac{\partial \mu }{\partial T}\right)}_{n}=-{\left(\frac{\partial S}{\partial n}\right)}_{T},$$
similarly to the “macroscopic” expression Equation (Equation 3). More importantly, we find the relation between the chemical potential and entropy from $\mu ={\left(\partial \bar{F}/\partial \bar{N}\right)}_{T}$, leading to:(6)$$\varepsilon -\mu =T{\left(\frac{\partial S}{\partial n}\right)}_{T}.$$
Note that the entropy used above is the entropy of the quantum dot with a mean excess population *n*, not the entropy of one of the charge states. Next, we derive the expression of this entropy. Since the quantum dot is an open system—it can exchange both energy and particles with the environment and, in principle, an infinite number of microstates is accessible to it—we use the Gibbs entropy expression.

In the steady state, at any point of time the quantum dot exists in one of the available charge states and in one of the microstates corresponding to each of the charge macrostates. The value of entropy has to represent both macrostate and microstate uncertainty—the uncertainty in the charge state of the quantum dot, and uncertainty in which microstate of each charge state is occupied.

We introduce a theorem: if a system can occupy *m* macrostates with probabilities of occupation $p_{i}$ and each macrostate in turn has $m_{i}$ microstates with probabilities $p_{ij}$, the total Gibbs entropy of this system is
(7)$$S=S_{c}+\sum_{i}p_{i}S_{i}$$
where $S_{c}$ is the “coarse” Gibbs entropy of macrostate occupation $S_{c}=-k_{B}\sum_{i}p_{i}\ln p_{i}$ and $S_{i}$ are the Gibbs entropies of the microstates: $S_{i}=-k_{B}\sum_{j}p_{ij}\ln p_{ij}$.

In the case of the quantum dot with two macrostates with the probabilities $p_{N+1}=n$ and $p_{N}=1-n$ corresponding to charge states with N+1 and *N* electrons, the entropy is equal to:(8)$$\begin{array}{c} S=-k_{B}\left(n\ln n-(1-n)\ln(1-n)\right)+\\ +nS_{N+1}+\left(1-n\right)S_{N}. \end{array}$$
Substituting the above, and the entropy of a two-macrostate system into Equation (Equation 6), we arrive at:(9)$$\frac{\varepsilon -\mu }{T}=k_{B}\ln\frac{1-n}{n}+\Delta S$$
where ΔS is the entropy difference between the two charge states $S_{N+1}-S_{N}$ and *n* is the mean excess population of the quantum dot. Like the initial Maxwell relation, the derivation of which it loosely follows, this equation holds true for any value of ε.

### 4.2. Applications and Experimental Evidence: A Two-Fold Degenerate Energy Level

We have arrived at Equation (Equation 9) directly from the corresponding Maxwell relation without any assumptions about the properties of electronic structure in the quantum dot. It connects the entropy difference between the two charge states with the energy level of the dot, the temperature and the mean excess population of the dot as shown in Figure 3.

First, we will show that the *general* thermodynamic relation describes the entropy measurements based on charge and conductance [15,17]. Charge state measurements [15] monitor the shift of ε−μ for the charge degeneracy point, where the probabilities of finding the system in both charge states are equal, as a function of temperature. At the charge degeneracy point n=1/2, the “coarse” entropy term $k_{B}\ln\left[\left(1-n\right)/n\right]$ in Equation (Equation 9) is equal to zero, reducing the equation to $\varepsilon_{1/2}-\mu =T\Delta S$, where $\varepsilon_{1/2}$ is the value of ε for the charge degeneracy point. The temperature-dependent energy for n=1/2 is shown in Figure 3 for charge state transitions where ΔS=0, $k_{B}\ln2$, and $-k_{B}\ln2$, corresponding to $d_{N+1}/d_{N}=1/1$, 1/2, and 2/1.

Next, we show from microscopic considerations (see Appendix A) that the peak in conductance corresponds to the “inverse non-degenerate” quantum dot population: n=1−f(ε) (for non-degenerate quantum dot in contact with a reservoir n=f(ε)). For the conductance peak, Equation (Equation 9) takes the form:(10)$$\frac{\varepsilon_{p}-\mu }{T}=k_{B}\ln\frac{f(\varepsilon_{p})}{1-f(\varepsilon_{p})}+\Delta S$$
which results in $\varepsilon_{p}=T\Delta S/2$, agreeing with both the theoretical evaluation [17] for the charge transport measurement setup and the experimental result of conductance peak shifting by $\pm k_{B}T\ln2/2$ in [8,18] for a two-fold degenerate level in a quantum dot. As shown in Figure 3, for ΔS=0 the conductance is maximum at n=1/2, while for $\Delta S=k_{B}\ln2$ and $-k_{B}\ln2$ the population at the conductance peak is $n_{p}=\sqrt{2}/\left(1+\sqrt{2}\right)$ and $n_{p}=1/\left(1+\sqrt{2}\right)$, respectively.

It is not unexpected that the two previously described entropy measurement methods are described by Equation (Equation 9), since it is simply a thermodynamic relation that holds for a system coupled to a bath close in (or close to) equilibrium. However, the fact that they are merely cases with specific values of *n* in a continuum, allows the determination of entropy for any fixed value of the mean excess population *n*. This is particularly useful for systems where due to limited gate control, not all values of *n* are accessible [27]. Moreover, Equation (Equation 9) can be used in reverse to find the dependence of occupation probabilities of two charge states of known dynamics (known entropy difference) on the gate voltage of the device without relying on rate equations, or when it cannot be determined, for instance when the levels and degeneracies are known, but not the hopping rates—see Appendix B.

### 4.3. A Single N-Fold Degenerate Energy Level

The general thermodynamic relation can be applied to systems with a higher degeneracy of the transport level, for instance, molecules with spatial symmetry that leads to extra spacial degeneracy for each charge state. A common example of such high-symmetry molecules are fullerenes [27,28], which have a five-fold degenerate HOMO (highest occupied molecular orbital) and a three-fold degenerate LUMO (lowest unoccupied molecular orbital).

Figure 4 shows the dependence of population and conductance on the reduced dot level energy $\left(\varepsilon -\mu \right)/k_{B}T$ for different values of $d_{N+1}/d_{N}$. For a transition between a $d_{N+1}$ degenerate state and a $d_{N}$ degenerate one, the entropy difference is equal to $\Delta S=k_{B}\ln d_{N+1}-k_{B}\ln d_{N}=k_{B}\ln\left(d_{N+1}/d_{N}\right)$. As expected, the reduced energy for the charge degeneracy point n=1/2 is $\Delta S/k_{B}=\ln\left(d_{N+1}/d_{N}\right)$ and the conductance peak energy is $S/2k_{B}=\ln\left(d_{N+1}/d_{N}\right)/2$.

## 5. General Thermodynamic Relation

### 5.1. Systems with Excited States

Now, we look at a more general system. Each charge state $N^{\prime }$ has a family of excited states with energies $E\left(N\right)+\delta \varepsilon_{i}$, where δε can be arbitrarily large. Each of the excited states can have its own degeneracy.

It is both usual and practical to define ε as the energy difference between the ground states of the charge state families [21]. We also define the additional mean energy of the charge state:(11)$${\bar{E}}_{N^{\prime }}=\sum_{i}p_{i}\delta \varepsilon_{i}$$
where the sum is over all the states corresponding to the charge state and $p_{i}$ is the probability of occupation of the microstate (for the ground state δε=0).

To write the mean free energy of the system, we need to include the mean additional energies of the charge states:(12)$$\bar{F}=E\left(N\right)+n\varepsilon +n{\bar{E}}_{N+1}+\left(1-n\right){\bar{E}}_{N}-TS$$
Following the derivation in Section 4.1, we arrive at a new form of the thermodynamic relation:(13)$$\frac{\varepsilon -\mu }{T}=k_{B}\ln\frac{1-n}{n}+\Delta S+\frac{{\bar{E}}_{N}-{\bar{E}}_{N+1}}{T}$$

If we define $\tilde{\varepsilon }$ as the difference between the mean energies of the charge states, $\tilde{\varepsilon }=E\left(N+1\right)-E\left(N\right)+{\bar{E}}_{N+1}-{\bar{E}}_{N}$, and Equation (Equation 13) can be simplified as:(14)$$\frac{\tilde{\varepsilon }-\mu }{T}=k_{B}\ln\frac{1-n}{n}+\Delta S$$
Figure 5 schematically shows the distinction between $\tilde{\varepsilon }$ and ε in the case of a multi-level system, its relation to the mean energies and the absence of this distinction in a system without excited states.

The final result (Equation (Equation 14)) could be obtained from the beginning, as the energy introduced with the additional population of the quantum dot is equal to $\tilde{\varepsilon }n$; however, $\tilde{\varepsilon }$ is harder to determine, both experimentally and computationally than ε. We should also note that the difference between the two parameters ε and $\tilde{\varepsilon }$ disappears in most experimental realisations of the measurement technique [8,18], since dε/dT is measured.

### 5.2. Discussion

Equation (Equation 14) is equivalent to a Maxwell relation Equation (Equation 3) written for a quantum dot. Therefore, it makes no assumptions about the “nature” of the entropy of the system—the physical origin of the microstate probabilities and energies and the tunnelling rates into each of the microstates, and can be applied to systems with all kinds of dynamics. The changes we propose to the thermodynamic approach when using it on a Coulomb-blocked device, are one of the main results of this work, and the system-independence follows from the thermodynamic origins of the approach.

Hartman et al. [15] performed an experiment comparing the dependence of the thermal shift of the charge degeneracy point on the magnetic field with the theoretical expression for the entropy of a single spin in a magnetic field: $S=k_{B}\left(p_{↑}\ln p_{↑}+p_{↓}\ln p_{↓}\right)$, where $p_{↑\;↓}={\left(1+\exp\left(\pm g\mu_{B}B/k_{B}T\right)\right)}^{-1}$. We note that while the excellent agreement between the two is not coincidental, the theoretical proof of the method given from detailed balance only applies to systems described by integer level degeneracies $d_{N+1}/d_{N}$.

To further justify the thermodynamic approach, we have derived the main result of the paper Equation (Equation 14) from microscopic considerations, starting from the Gibbs distribution (see Appendix C) for a system with excited states, to show that the “top-down” thermodynamic approach agrees with the more standard “bottom-up” microscopic one, common in the field. This also serves as evidence for the validity of our choices of entropy and chemical potential for the problem.

It may seem that the thermodynamic approach we suggest is simply a reformulation of the rate equation and, knowing the system, one can always find the shift in the charge or conductance traces. However, as the thermodynamic approach produces the value of entropy without making any prior assumptions, it can provide an important tool for choosing a physical model for an unknown system.

One important note, however, is that while we have shown that the charge state measurement method is applicable to all conceivable systems, our proof of the validity of the method based on conductance relies on the assumption that the energy spacing between the levels corresponding to each of the charge states is small—the hopping rates to all of them are equal (see Appendix A). This suggests that the applicability of the conductance measurement is narrower than that of the charge state measurement.

## 6. Conclusions

We present a framework for using classical thermodynamic relations for small few-electron systems, which involves using non-standard in the field definitions for entropy and chemical potential. Using it, we derive a thermodynamic relation applicable to nanodevices in the state of Coulomb blockade, without any additional prior assumptions about the system. Equation (Equation 14) describes both previously proposed entropy measurement methods as special cases, based on what mean excess dot population *n* has been used as the experimental benchmark. The fact that our equation describes the behaviour of the system under a continuum of external parameters allows to broaden the experimental scope of the methods, for instance, by measuring the entropy of a system where mean population is known, but a charge-degeneracy state is not accessible.

Additionally, we discuss the form of the relation for complex systems with multiple excited states with different degeneracies and large level spacings for each charge state and prove it independently in this case from microscopic considerations, as futher evidence towards our choice of thermodynamic parameters.

Our approach demonstrates the subtlety of applying thermodynamic relations to microscopic systems and its agreement with previous results obtained by different methods, both theoretical and experimental results indicate that our application and choice of parameters is correct. Thus, we can hope that we have provided a simple framework for applying thermodynamic relations to common experimentally studied nanodevices, which can be expanded for use with other microscopic systems with more than two charge states or detectable macrostates of a different origin using the same toolkit of the mean population, thermal bath-defined chemical potential and entropy that includes both microstate and charge state uncertainty.

## Figures and Tables

**Figure 1 entropy-23-00640-f001:**
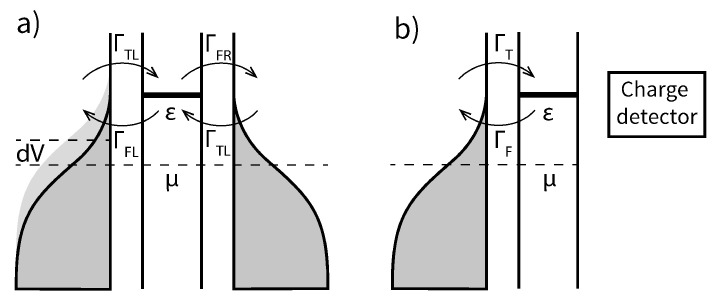
Experimental regimes of Coulomb-blocked nanodevices: (**a**) A quantum dot coupled to a thermal bath and exchanging electrons with it. The charge state of the quantum dot can be independently determined. (**b**) A quantum dot coupled to two electrodes through tunnel junctions. A potential difference dV between can be applied between them and current through the quantum dot is measured.

**Figure 2 entropy-23-00640-f002:**
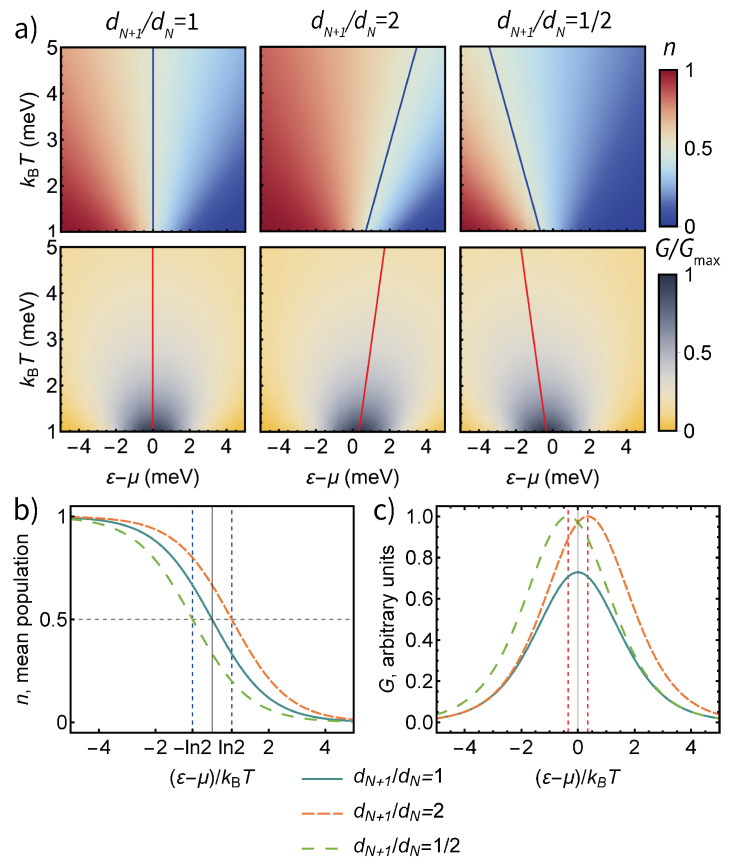
(**a**) The dependence of the mean excess population and conductance of a quantum dot coupled to a heat bath for a non-degenerate transport level and a two-fold degenerate one with even and odd *N*, respectively—$d_{N+1}/d_{N}=$1, 2, 1/2. The blue lines show the charge degeneracy point n=1/2, and red lines the conductance peak for each temperature. (**b**) The dependence of mean excess population on the dimensionless energy parameter $\left(\varepsilon -\mu \right)/k_{B}T$ for a non-degenerate transport level and a two-fold degenerate one with even and odd *N*, respectively. (**c**) The dependence of the conductance of a quantum dot on the dimensionless energy parameter $\left(\varepsilon -\mu \right)/k_{B}T$ for a non-degenerate transport level and a two-fold degenerate one with even and odd *N*, respectively.

**Figure 3 entropy-23-00640-f003:**
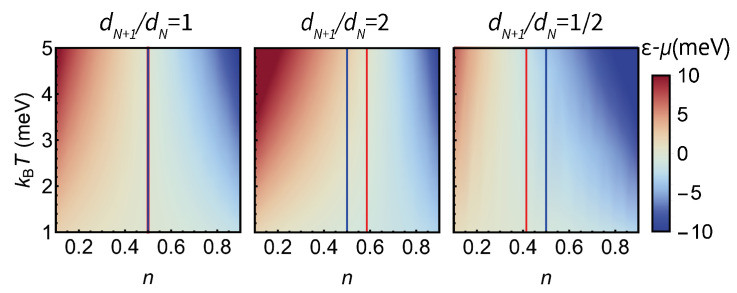
Illustration of Equation (Equation 9)—the dependence of the dot energy level on the population and temperature for a non-degenerate dot energy level, $d_{N+1}/d_{N}=2$ and $d_{N+1}/d_{N}=1/2$, respectively. The blue line shows the space of $\varepsilon_{1/2}$ and the red line $\varepsilon_{p}$—corresponding to the conductance peak.

**Figure 4 entropy-23-00640-f004:**
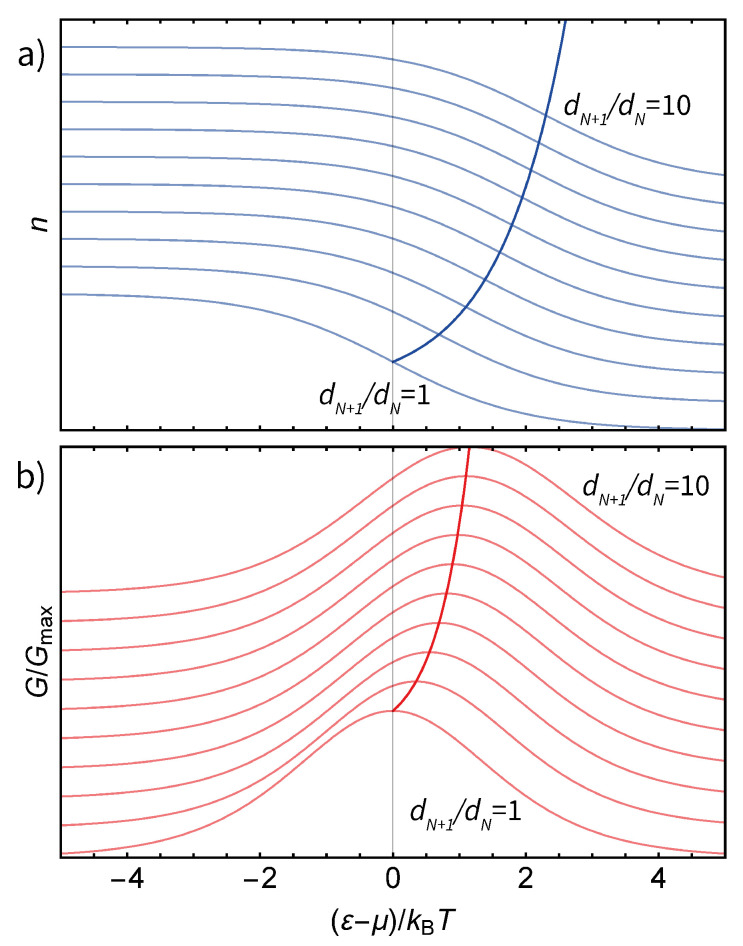
(**a**) The dependence of the population of a quantum dot on $\left(\varepsilon -\mu \right)/k_{B}T$ for different level degeneracies. The inflection points n=1/2 fall on the exponential curve, as predicted. (**b**) The dependence of the conductance of a quantum dot on $\left(\varepsilon -\mu \right)/k_{B}T$ for different level degeneracies. The conductance peaks fall on the exponential curve with a twice greater argument.

**Figure 5 entropy-23-00640-f005:**
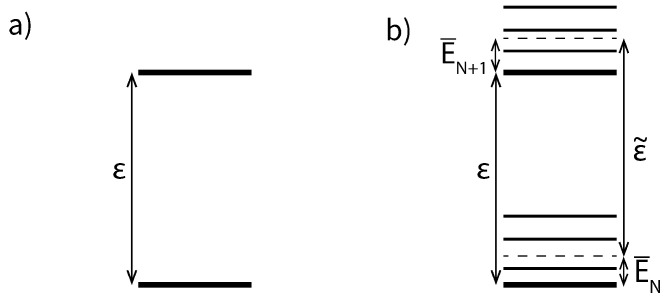
(**a**) The energy level structure of a quantum dot if all states with the same charge are energetically degenerate. (**b**) Energy level structure in a quantum dot with excited states. Each charge state has a family of excited states at energies $E\left(N^{\prime }\right)+\delta \varepsilon_{i}$ and there is a non-zero mean energy ${\bar{E}}_{N^{\prime }}$ of the excited states above $E_{N}$. ε is the energy difference between the ground states, while $\tilde{\varepsilon }$ is the difference between the mean energies of the charge states.

## Data Availability

Not applicable.

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
