# Peer review of "A Thermodynamic Approach to Measuring Entropy in a Few-Electron Nanodevice"

_entropy, 2021, doi:10.3390/e23060640_

Round 1

Reviewer 1 Report

The paper discussed the new approaches to measuring the entropy of a mesoscopic system, one (Ref. 15) using the Maxwell relation, and the other (Ref. 17) using thermoelectric measurements. The authors claim to have generalized these approaches  and show that they are special cases of a more general approach. I could not find any support for these claims in the paper. First, about the generalization. The first approach is based on thermodynamic relations, and thus are applicable to any system in equilibrium. The second, as was shown in Ref. 17, is applicable to an arbitrary, multilevel interacting system weakly coupled to the reservoirs. I couldn't see any additional generalization in the paper, just a bunch of examples where the approaches actually work. Second, I was very excited to read in the abstract that they are going to show a formalism where these two approaches appear as special cases. I couldn't find such a formalism in the paper. 

To summarize - if the authors stand by the claims they have made in the abstract, they should be clear exactly where these claims are substantiated. Special cases where the formalisms that have already been developed work are not examples of a generalization. I cannot recommend publication of the manuscript in its present form.

Reviewer 2 Report

In 2018 N. Hartmenn et al. (Ref. 15) published an article in
Nature Physics with the title: "Direct entropy measurement 
in a mesoscopic quantum system. They use a Maxwell relation
(Eq. (1)) to perform  an  entropy-to-charge  conversion.

The authors of the present manuscript, Eugenia Pyurbeeva and 
Jan Mol (EP and JM) revisit this derivation and argue that steps 
in it should be replaced by a slightly different approach better
suited for small systems and an analysis strictly adhering
to thermodynamics. 

In exactly this sense I consider the manuscript (by EP and JM) 
an important contribution to the difficult question of direct 
measurement of entropy in small systems. I thus recommend its 
publication in the mdpi journal Entropy.

The manuscript is well written and organized and all derivations
are thoroughly explained. 

The direct measurement of entropy in small systems is a very
important subject of intense theoretical and experimental 
value. Both approaches investigate the entropy change during
the change of electron number in the quantum dot between N
and N+1. Both approaches ignore the many-body structure of the 
electron system and the interactions between the electrons.
This fact leaves me with feeling of butterflies in my stomach,
but I acknowledge the statement of the authors (EP and JM) in the 
conclusion section:

   "Additionally, we provide proof that the result holds 
    true for much more complex systems than those that have 
    been considered before: systems with multiple excited
    states with different degeneracies and large level 
    spacings for each charge state.
    Our approach demonstrates the subtlety of applying 
    thermodynamic relations to microscopic systems and its 
    agreement with previous results obtained by different
    methods, both theoretical and experimental indicates that 
    our application and choice of parameters is correct."
